# Fabrication of Fe-Si-B Based Amorphous Powder Cores by Spark Plasma Sintered and Their Magnetic Properties

**DOI:** 10.3390/ma15041603

**Published:** 2022-02-21

**Authors:** Liang Yan, Biao Yan, Yin Jian

**Affiliations:** 1School of Intelligent Manufacturing and Control Engineering, Shanghai Polytechnic University, Shanghai 201209, China; yanliang@sspu.edu.cn; 2School of Material Science and Engineering, Tongji University, Shanghai 201209, China; 84016@tongji.edu.cn; 3School of Mechanical Engineering, Tongling University, Tongling 244000, China

**Keywords:** spark plasma sintering, mechanical ball milling, iron-based amorphous alloy, soft magnetic powder core, core loss

## Abstract

Mechanical ball milling was used to coat SiO_2_ nanopowder on a Fe-Si-B amorphous powder in this study. The Fe-Si-B/SiO_2_ core–shell amorphous composite powder was obtained after 6h of ball milling. At 490 °C, the amorphous powder is thermally stable. Discharge plasma sintering was used to create a Fe-Si-B/SiO_2_ magnetic powder core (SPS). At a sintering temperature of 420 to 540 °C, the phase composition and magnetic characteristics of the magnetic particle core were investigated. Scanning electron microscopy (SEM) and X-ray diffraction (XRD) were used to examine the structural features of the magnetic particle core. A precision resistance tester and a vibrating sample magnetometer were used to assess the resistivity and magnetic characteristics of the magnetic particle core. The findings showed that Fe_3_Si and Fe_2_B are the phases generated during spark plasma sintering. High-frequency power loss increases as density rises. However, at the measured frequency, the magnetic permeability of the magnetic particle core changes slightly and has excellent frequency characteristics, making it appropriate for use in high-frequency components.

## 1. Introduction

Due to their lack of grain boundary and magnetocrystalline anisotropy, such as having soft magnetic properties like high resistivity and low eddy current loss, Fe-based amorphous alloys have been widely used in high-frequency bands [1]. However, iron-based amorphous alloys, on the other hand, can only be treated by melting due to the high cooling rate required for the creation of the amorphous phase, which limits their technical applications [2]. In recent years, many alloy systems and amorphous alloys with the required shape and size have been prepared using powder metallurgy technology [3]. Different authors have proved the effectiveness of powder metallurgy technology in inducing amorphization in a variety of iron-based alloys, such as Fe-Si-B [4], Fe-Zr-B [5], Fe-Al-P-B-C [6], Fe-Co-Ni-Zr-B [7], Fe-Cu-Nb-Si-B [8], and Fe-Ni-P-Si [9]. The key to avoiding crystallization is to keep the preparation temperature of the amorphous magnetic particle core under control. Due to the lack of a substantial supercooled liquid zone, preparing an iron-based amorphous powder core remains difficult. Compared with traditional sintering methods, discharge plasma sintering (SPS) has significant advantages, such as short duration and fast heating rate, which can inhibit grain growth and the crystallization of the amorphous materials during sintering [10]. In recent years, large volume iron-based BGAs such as Fe-Cu-Ni-Mo-C [11], Ce-Fe-B [12], Fe-Si-B-Cu-Nb [13], and Fe-Cr-Mo-Y-B-C [14] have been successfully prepared by the SPS technology.

Using the core–shell structure composite powder created by mechanical ball milling, the gas-atomized Fe-Si-B amorphous powder and SiO_2_ nano core–shell composite powder were prepared in this research. SiO_2_ is chosen to be added to Fe-Si-B, which can exhibit high magnetic flux density, good soft magnetic properties, and low core loss (6 wt.%). These are added to the SiO_2_ insulator layer and form on the surface of the alloy powder so that it can effectively reduce the eddy current and, consequently, reduce the core loss. The effects of the spark plasma sintering process on the phase transformation, microstructure, and the magnetic properties of the magnetic powder core under different sintering temperatures and holding times were studied. The study establishes a theoretical basis for the research and development of high-performance amorphous alloy materials.

## 2. Experimental Materials and Methods

The raw materials used in the study include aerosolized Fe-Si-B amorphous powder (purity > 99%, 40 μm) and SiO_2_ powder (purity > 99%, 50 nm). The amount of SiO_2_ powder added is 6 wt.%. Fe-Si-B amorphous powder and SiO_2_ powder are added into the ball milling tank and milled for 6h under the protection of argon. The following are the ball milling process parameters: The ball milling medium is a 3 mm stainless steel ball with a 20:1–35:1 ball material ratio and a rotating speed of 200 rpm to 250 rpm. The core–shell composite powder is made with a Fe-Si-B core and a SiO_2_ shell. Subsequently, the composite powder is preloaded with an SPS (Sumitomo sps-3.20mk-iv, Shanghai, China) graphite model and put into the SPS sintering furnace (SP Shanghai Dongyang Carbon Co., Ltd. S-3.20 mk-IV, Shanghai, China). The process parameters are: sintering pressure 40 MPa, holding time 1–5 min, and sintering temperature 420~540 °C.

The structure of the powder and powder core were identified by X-ray diffractometer (dx-2007, China Dandong Fangyuan Co., Ltd. Dandong City, Liaoning Province, China. 30 KV and 30 mA cu-k α). The morphology and the local chemical uniformity of the powder and the powder core were studied using the scanning electron microscope (SEM) (Nova Nanosem 450, Fei, Portland, OR, USA). and the energy dispersive spectrometer (EDS) (ultra, EDAX, Washington, DC, USA). The static magnetic properties of the dense magnetic powder core were measured using the vibrating sample magnetometer (VSM) (American quantum design company, Beijing, China). The loss and permeability of the magnetic particle core at 20 MT were measured by a soft magnetic AC measuring instrument (mats-2010sa/500 K, Linkioin, Loudi, China) in the range of 1 kHz to 90 kHz.

## 3. Results and Discussion

### 3.1. The Microstructure and the Phase Composition of Fe-Si-B/SiO_2_ Composite Powder

Mechanical ball milling was used to generate Fe-Si-B powder-coated nano SiO_2_ powder with a core–shell structure in this work. Figure 1 depicts the SEM photos of the Fe-Si-B powder before and after being coated with the nano-SiO_2_ powder. The figure shows that the Fe-Si-B powder presents a typical gas phase atomized powder morphology and is spherical and smooth before being coated with SiO_2_ (Figure 1a). The plastic deformation generated by the impact and extrusion during the continuous mechanical ball milling process resulted in an uneven shape of the composite powder and a rather rough surface (Figure 1b). The particle size distribution diagram of the core–shell powder structure is shown in Figure 2. The particle size distribution of the powder exhibits a single peak with a median particle size and average particle size of 30.47 μm and 25.44 μm, respectively. This is consistent with the average particle size observed under the scanning electron microscope. Figure 3 shows the XRD patterns of the Fe-Si-B powder and the Fe-Si-B/SiO_2_ composite powder, respectively. There is no distinguishable crystal diffraction peak in each XRD pattern, which is an indication that they are completely amorphous structures.

The DSC curve of the Fe-Si-B/SiO_2_ powder is shown in Figure 4, and it reveals that the DSC curve is a typical Fe-based amorphous curve with a glass transition peak and crystallization exothermic peak, and with a glass transition temperature (T_g_) of 440 °C, an initial crystallization temperature (T_x_) of 480 °C, and a supercooled liquid region of 440–480 °C—with a width of 40 °C. The crystallization peak temperature (T_p)_ is about 550 °C. In the supercooled liquid region, the powder has large viscous flow, atomic diffusion, and superplasticity [15], and, consequently, the sintering temperature is selected in this range during the experiment. Due to the temperature differential between the mold and the core powder in the SPS sintering, the sintering temperature is 460 °C to prevent the sample from crystallizing.

### 3.2. Microstructure of the Fe-Si-B/SiO_2_ Magnetic Particle Core

Based on the DSC curve results of the Fe-Si-B/SiO_2_ composite powder, the designed SPS sintering pressure is 40 MPa with a holding time of 1 min, and the sintering temperature range is 420–540 °C. Figure 5 shows the XRD pattern. The results showed that with an increase in the sintering temperature, α—the crystal phases of Fe (Si), Fe_3_Si, and Fe_2_B—will show obvious diffraction peaks.

Figure 6 depicts the relative density of the sintered blocks at various sintering temperatures, thereby demonstrating that the density increases with an increase in the sintering temperature. The relative densities were 93.2%, 98.31%, 98.72%, and 98.93%, respectively. When the sintering temperature is lower than 440 °C, the block maintains an amorphous structure and crystallization occurs when it reaches 480 °C [16]. It is only by sintering in the supercooled liquid region and promoting the diffusion and mass transfer at the powder particle interface that we can obtain a fully dense alloy [17].

The SEM image of the fracture of the magnetic particle core sintered at 460 °C is shown in Figure 7. There are many apparent remnants of the liquid layer on the surface of the particle and between the particles, and the SiO_2_ nanopowder melts and fills the gap in the Fe-Si-B particles. This shows that plasma formation is highly exothermic in the SPS process, hence, the temperature at the edge of the surface can rise to thousands of degrees Celsius [18]. This phenomenon is characterized by the formation of sparks between the surfaces of the opposite particles above the gap. There are many gaps and spaces between the particles as shown in Figure 7b, which result in the accumulation of charges during the SPS process [19].

When preparing compacts with SPS, the potential pollution of graphite punch-and-die carbon to the sample must be considered [20]. In order to confirm the carbon contamination of the punches and dies from the SPS equipment, an EDX microanalysis was performed on the edge of the compact slice obtained by SPS at 460 °C—as shown in Figure 8. It is found that there is a large amount of carbon on the surface of the magnetic particle core, and the diffusion depth of carbon is estimated to be about 2–4 μm. Figure 9 shows the EDS spectrum of the polished surface of the Fe-Si-B/SiO_2_ powder core. The results show that iron is fairly evenly distributed in the analysis area. The distribution of silicon and boron is also similar, although these two chemical elements do not exist in both compounds (Fe-Si and Fe-B). This can be explained by the uniform distribution of Fe-Si and Fe-B phases in the sintered billet.

Compared with an Fe-Si-B ribbon, the Fe-Si-B magnetic powder core has a stronger demagnetization due to its annular shape. Therefore, a stronger magnetic field is required to achieve magnetic saturation. Figure 10 shows an Fe-Si-B ribbon, Fe-Si-B powder, and Fe-Si-B/SiO_2_ magnetic particle core at different sintering temperatures. It can be seen from the figure that the Fe-Si-B ribbon, Fe-Si-B powder, and Fe-Si-B/SiO_2_ magnetic particle core show a typical soft magnetic circuit.

Compared with ribbon samples, stronger fields should be required for the magnetic saturation of a power disc sample due to a stronger demagnetization factor resulting from the disc shape. However, these powder discs also exhibited a typical soft magnetic loop and the magnetization saturated at the magnetic field at about 160 kA/m. The effects when the magnetization is at 800 A/m (B_800_) and μ_e_ is at 1 kHz on the resistivity and Hc external magnetic fields are shown in Table 1. The table lists the data compared with the amorphous ribbon. When the sintering temperature increases from 460 °C to 540 °C, the magnetization of the magnetic powder core increases from 1.15 T to 1.54 T, because the increase of temperature increases the density of the sample and makes its Ms gradually close to the theoretical value. In addition, the Hc value of the amorphous ribbon is very different from that of the magnetic powder core. This is because the magnetic powder core is made of powder pressed together by SPS and sintered at a low temperature. Furthermore, there is a great stress in the interior, thereby leading to an increase of coercivity. In addition, the magnetic powder core may also have small local defects after sintering, which result in gaps between the powders. Thus, the μ_e_ decreases from 6700 to about 50 and Hc increases from 7.9 A/m to more than 800 A/m. For saturation magnetization, the original powder has the highest Ms of 1.54 T, which is close to the Ms of the GB standard 1k101Fe-Si-B amorphous ribbon. The resistivity of the Fe-Si-B/SiO_2_ decreases with the increase of sintering temperature, but it is significantly higher than that of the comparative amorphous belt sample I (137.1 µΩ·cm), which is because the SiO_2_ insulating layer hinders the electron movement in Fe-Si-B particles and achieves a good insulation effect.

Figure 11 shows the variation curves of the initial relative permeability and the core loss of sintered samples within the frequency range of 10–100 kHz with a maximum induction field of B_m_ = 0.05 T. The maximum loss in the high-frequency spectrum is a 5 min holding time. Sintering times of less than 5 min can reduce the magnetic loss of the sample. The density and resistivity characteristics of the magnetic particle core helps explain the evolution of the magnetic properties. Generally, in the high-frequency range, the eddy current loss is directly proportional to the square of the applied frequency and is inversely proportional to the compact resistivity. Comparing the initial relative permeability of the sintered samples revealed that the sample with the highest permeability is the one kept for 5 min, which is almost twice as large as that obtained by sintering for 1 min. Permeability is very sensitive to density, microstructure, and purity, given that the XRD, SEM, and EDS results assumed that the purity and microstructure are similar. As a result, it can be inferred that the main cause of the change in permeability is density. The increase in density causes a direct rise in the initial relative permeability. The sample’s density is low, and its porosity is high, resulting in poor magnetic permeability. During the 5 min holding period, the permeability is essentially consistent over the whole frequency range.

Figure 12 shows the effect of the sintering temperature on the AC magnetic properties of the sample. The results showed that the effects of the sintering temperature on the permeability and the magnetic loss are the same. The initial relative permeability is enhanced by increasing the sintering temperature. In addition, when the sintering temperature is increased, the iron loss of the sample increases, but when the sintering temperature is 540 °C, the iron loss decreases. Conversely, at high frequency, the loss of amorphous alloy is mainly eddy current loss. Also, the grain size increases with an increase in sintering temperature, which increases the effective radius of the eddy current, and thereby increases the energy required for the domain rotation and the magnetization or demagnetization—resulting in an increased loss. Furthermore, when the sintering temperature exceeds the crystallization temperature, the amorphous phase crystallizes completely and the bulk is dominated by nanocrystals. In this process, the density of the bulk increases and the micro-stress in the powder is released during the sintering process. Their combined effect reduces the force hindering the movement of the domain wall and domain rotation in the material [21], resulting in the reduction of the eddy current loss by the material.

## 4. Conclusions

SMCs with the micro-cell structure were prepared using the SPS sintering spherical atomized Fe-Si-B amorphous powder coated with SiO_2_ nanopowder.
(1)The core–shell composite powder of the Fe-Si-B coated nano SiO_2_ nanopowder was prepared by mechanical ball milling. The width of the supercooled liquid region of the composite powder is up to 40 °C, and the amorphous alloy has a strong amorphous forming ability.(2)The Fe-Si-B/SiO_2_ bulk amorphous nanocrystalline magnetic materials were prepared by spark plasma sintering. When the sintering pressure was 40 MPa, the sintering temperature range was 420–540 °C with a holding time of 1 min, and the prepared bulk Fe-Si-B particles (core) were well separated and insulated by the SiO_2_ (shell) intergranular layer in the core of the powder—and the density reached 98.93%. The block crystallizes at about 460 °C, and its crystalline phases are Fe_3_Si and Fe_2_B.(3)When the sintering and holding times are increased, there is an improvement in the density, maximum relative permeability, and magnetic loss of the Fe-Si-B/SiO_2_ sintered block. The magnetic permeability of the Fe-Si-B/SiO_2_ sintered block is stable in the high-frequency region. The magnetic properties are best when the sintering temperature is 420 °C for 2 min, with excellent soft magnetic properties.

## Figures and Tables

**Figure 1 materials-15-01603-f001:**
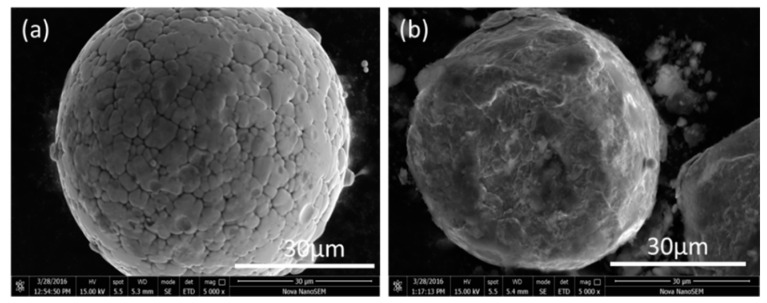
SEM of: (**a**) Fe-Si-B powder and (**b**) Fe-Si-B/SiO_2_ composite powder.

**Figure 2 materials-15-01603-f002:**
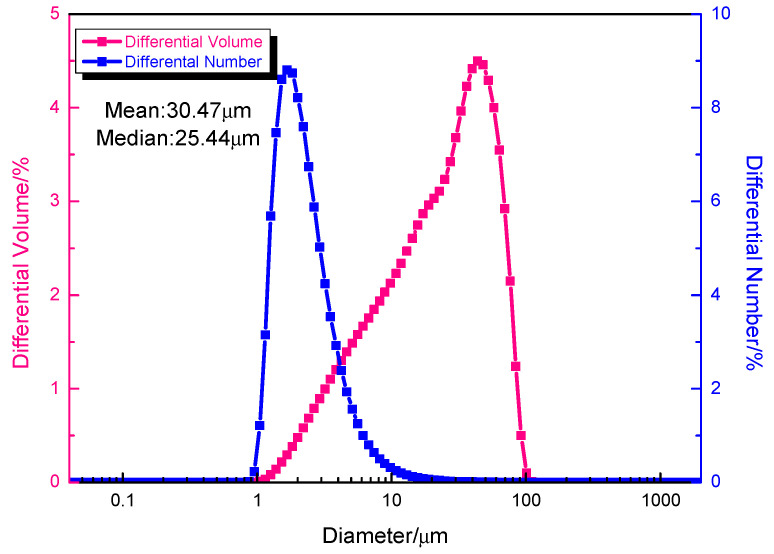
Particle size distribution of Fe-Si-B/SiO_2_ composite powder after grinding.

**Figure 3 materials-15-01603-f003:**
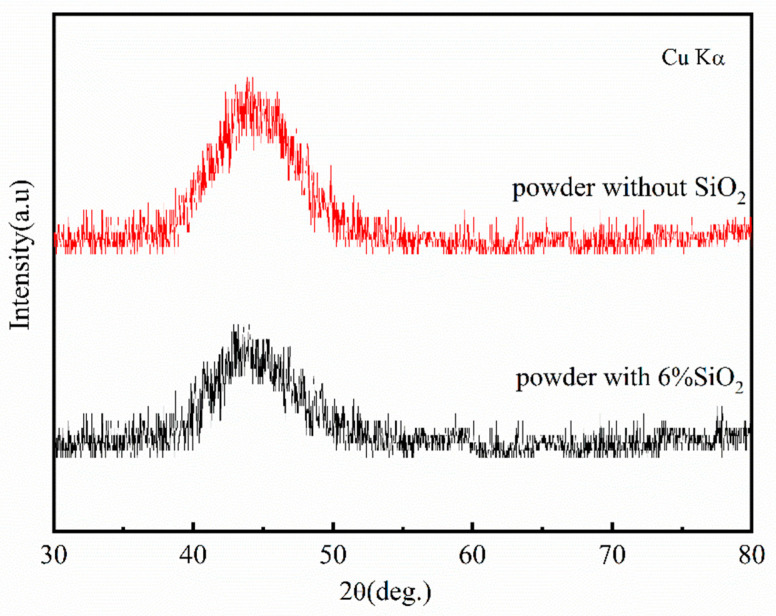
XRD of Fe-Si-B powder and Fe-Si-B/SiO_2_ composite powder.

**Figure 4 materials-15-01603-f004:**
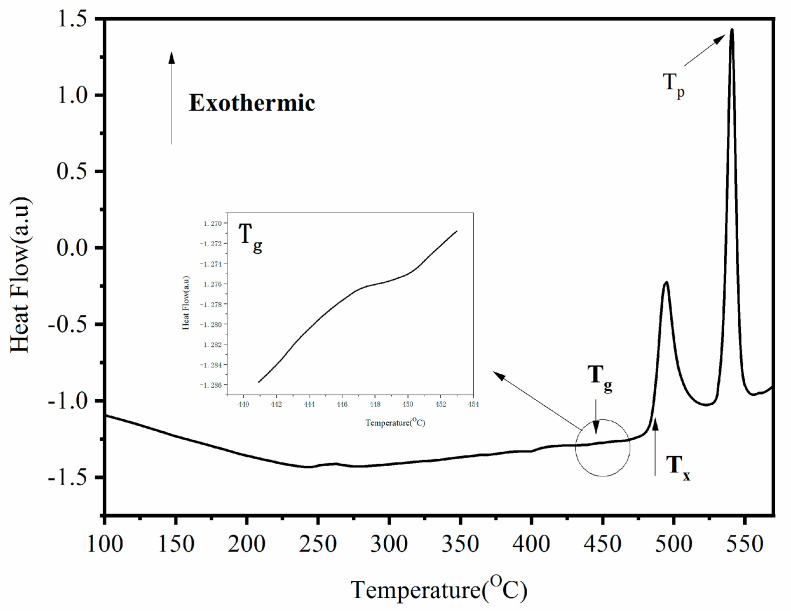
DSC Curve of Fe-Si-B/SiO_2_ powder.

**Figure 5 materials-15-01603-f005:**
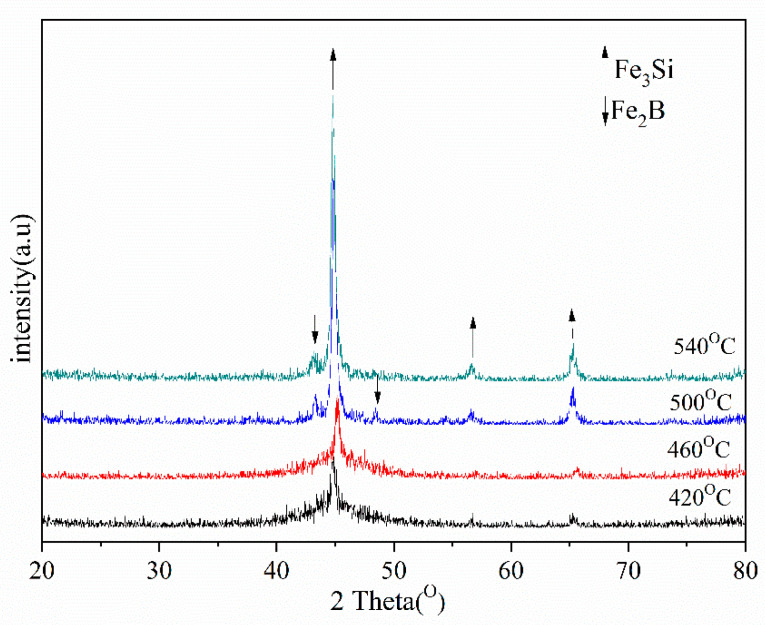
XRD of Fe-Si-B/SiO_2_ magnetic powder core at different sintering temperatures.

**Figure 6 materials-15-01603-f006:**
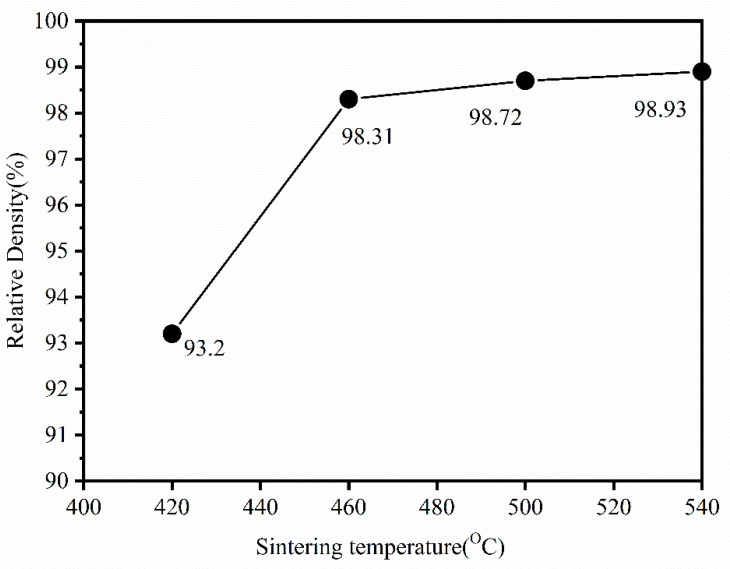
Relative density of Fe-Si-B/SiO_2_ magnetic particle core at different sintering temperatures.

**Figure 7 materials-15-01603-f007:**
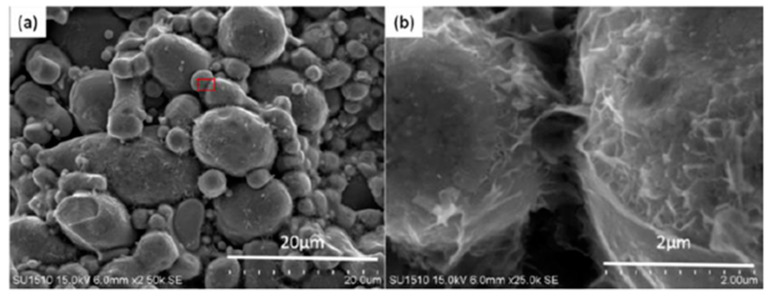
SEM image of the fracture of Fe-Si-B/SiO_2_ composite sintered at 480 °C (**a**), (**b**) is the SEM enlarged image of the selected area (**a**).

**Figure 8 materials-15-01603-f008:**
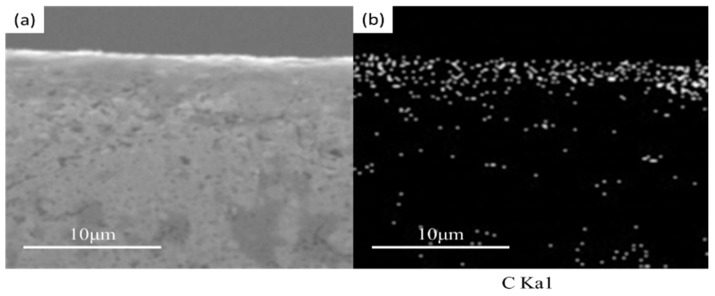
Distribution maps of the carbon elements obtained by X-ray microanalysis on the edge of the compact obtained by spark plasma sintering at 460 °C. (**a**) SEM image of compact; (**b**) Elemental mapping C.

**Figure 9 materials-15-01603-f009:**
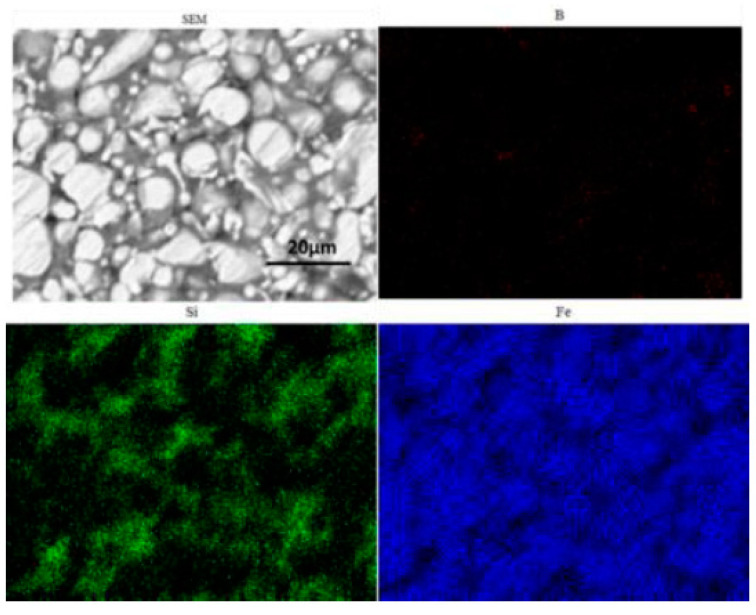
EDS of polished surface of Fe-Si-B/SiO_2_ magnetic particle core.

**Figure 10 materials-15-01603-f010:**
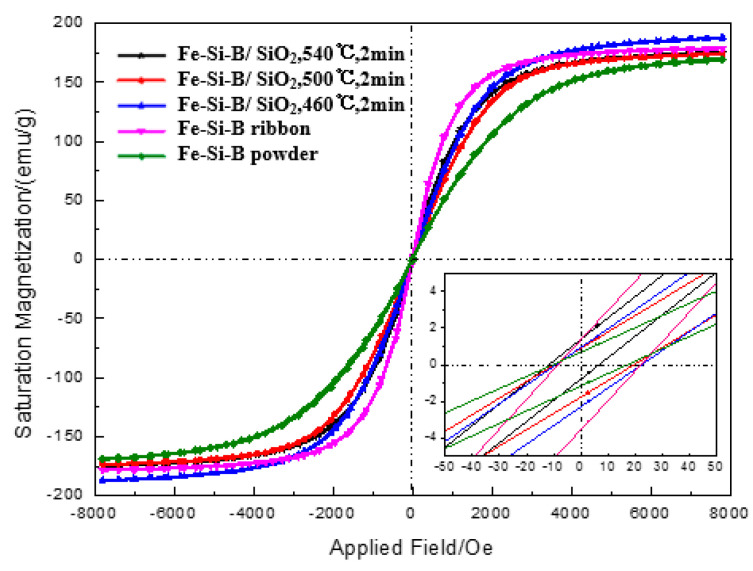
B-H hysteresis loop of an Fe-Si-B ribbon, Fe-Si-B powder, and Fe-Si-B/SiO_2_ magnetic particle core at different sintering temperatures.

**Figure 11 materials-15-01603-f011:**
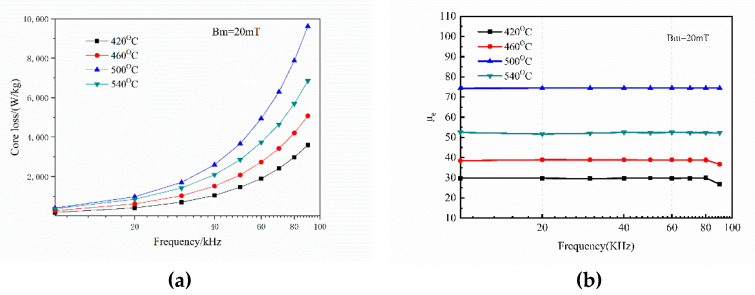
Effect of holding time on AC magnetic properties of Fe-Si-B/SiO_2_ magnetic particle core. (**a**) the core loss (**b**) the initial relative permeability.

**Figure 12 materials-15-01603-f012:**
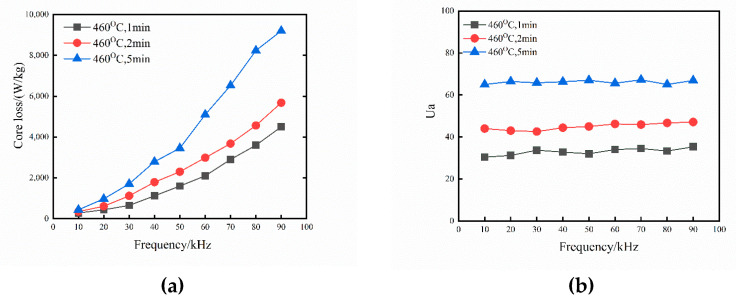
Effect of sintering temperature on AC magnetic properties of Fe-Si-B/SiO_2_ magnetic particle core. (**a**) the core loss (**b**) the initial relative permeability.

**Table 1 materials-15-01603-t001:** The magnetization under the applied field of 800 A/m (B_800_), μ_e_ at 1 kHz and Hc.

	B_800_ (T)	Hc (A/m)	μ_e_	Resistivity (µΩ·cm)
**Fe-Si-B ribbon**	1.56	7.9	6700	137.1
**Fe-Si-B powder**	1.54	2111	590	-
**Fe-Si-B/SiO_2_, 460 °C, 2 min**	1.15	934	45	691.44
**Fe-Si-B/SiO_2_, 500 °C, 2 min**	1.23	809	48	449.42
**Fe-Si-B/SiO_2_, 540 °C, 2 min**	1.54	3574	76	301.56

## Data Availability

Not applicable.

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
