# Peer review of "Fabrication of Fe-Si-B Based Amorphous Powder Cores by Spark Plasma Sintered and Their Magnetic Properties"

_materials, 2022, doi:10.3390/ma15041603_

Round 1
Reviewer 1 Report
Article is devoted to the development of new soft magnet materials by SPS of the composite FeSiB-SiO2 powders. Experimental design is explained appropriately, but conclusions from the experiments have to be discussed in more detail.
- It is stated that SiO2 powder melts during SPS due to plasma formation. First of all, plasma formation during SPS is unlikely because of low voltage of this process (D.M.Hulbert et al. The absence of plasma in spark plasma sintering, J.Appl. Phys. 104, 033305, 2008). Local transient contact melting of powder particles is possible, but melting temperature of the FeSiB alloy is much lower than SiO2 and it is alloy melting that is expected. In relation to this topic, Ref. 19 to the Journal of Integrated Agriculture looks strange.
- According to the Table 1, magnetic properties of the obtained material are drastically different from the properties of the FeSiB amorphous ribbons. In my opinion, it should be explained why this material is still considered as "excellent soft magnet"
- There are plenty of misprints in the manuscript
Author Response
Thank you for your advice,
1.I read the relevant literature and learned that: in the SPS sintering process, the pulse current causes the discharge effect between particles, increases the local temperature, leads to local melting on the particle surface, and the oxide on the particle surface peels off; At the same time, the discharge effect between particles can purify the particle surface, remove impurities (such as surface oxides) and adsorbed gases on the particle surface, and realize rapid sintering. Therefore, my expression” the plasma formation is highly exothermic” Change to” The pulse current causes the discharge effect between particles, increases the local temperature。”
2.To” Local transient contact melting of powder particles is possible, but melting temperature of the FeSiB alloy is much lower than SiO2 and it is alloy melting that is expected.”, My explanation is that because I use SiO2 coated FeSiB composite powder, and the coating thickness reaches 2 microns, because the pulse current of spark plasma sintering occurs at a high frequency, the electric discharge generated at the non-contact part of composite powder particles and the Joule heat generated uniformly at the contact part of composite powder particles promote the activation and diffusion of powder particle atoms, The diffusion coefficient is higher than that of the samples prepared under normal hot pressing conditions, which makes the sintering of SiO2 powder faster;
3.To” it should be explained why this material is still considered as "excellent soft magnet": The MS of sintered samples increases with the increase of temperature. The increase of temperature increases the density of samples and makes its MS gradually close to the value of theoretical density. The saturation magnetization of the amorphous block with the highest density is 1.54T. Therefore, its magnetic properties are excellent

Reviewer 2 Report
The presented article for review is factually and technically correct. Research well selected and described, but the article is a very correct technical report. The authors do not indicate any potential applications. Correct conclusions. Poor quality drawings.
- What is the main question addressed by the research? Is it relevant and interesting?
The paper investigates the effect of the spark plasma sintering process on the phase change, microstructure and magnetic properties of the magnetic powder core at various sintering temperatures and holding times. The discussed topic is interesting and provides the basis for further research on this topic. - How original is the topic? What does it add to the subject area compared with other published material?
The topic is original, in paper the gas atomized FeSiB amorphous powder and SiO2 nano coreshell composite powder were prepared by using the core-shell structure composite powder prepared by mechanical ball milling. The subject add a new temperature range and heating time, which gives an improvement in the magnetic properties. - Is the paper well written? Is the text clear and easy to read?
The paper is well written and is clear and easy to read. - Are the conclusions consistent with the evidence and arguments presented? Do they address the main question posed?
Yes

Author Response
Thank the reviewers for their affirmation of my work. In view of the shortcomings of the paper, I have made modifications within my ability. Please review it.

Reviewer 3 Report
This manuscript reports on the dependence of various physical properties of FeSiB/SiO2 core-shell powder on precise synthesis conditions. I have a number of concerns about this manuscript:
- The quality of the figures is too low. While for some of the figures, this is not a big problem and the main content can still be deduced, for others (in particular Figs. 10 and 11) the legend is not readable. In Figs. 10 and 11 is is also unclear which the data correspond to the left and which to the right y-axes.
- There is no motivation why SiO2 is chosen to be added to FeSiB (has SiO2 been used in literature before?), or why 6 wt% are added.
- The abstract mentions resistivity and a precision resistance tester – but in the main text I find no corresponding results. Resistivity results would certainly be useful to have at hand to support e.g. the discussion about Eddy currents.
- Figure 2 shows particle size distribution. This is certainly useful to have – but it is completely unclear how these distributions were determined.
- Figure 3 shows XRD patterns and as the authors correctly state, there are no obvious sharp Bragg peaks characteristic of a significant crystalline volume fraction. However, there is an obvious (broad) peak between 40 and 50 deg – is this typical of these amorphous materials? What is the origin of this feature?
- DSC curve shown in Fig. 4: was this measured on heating or cooling? What is the most obvious peak (at around 540 C) due to and what is this arrow TC for?
- More importantly, also in Fig. 4 there is an assignment of a glass transition temperature to 440 C. Looking at the figure I see no obvious feature at this temperature. To make a convincing case, maybe an inset with the immediate region enlarged could be shown, along with a brief discussion with references that only a very small feature can be expected at Tg.
- How was the relative density determined? Probably by weighing and somehow determining the volume of specific samples. Still this information should be given as the density appears to be rather important for the overall conclusions.
- Referencing to Fig. 8 the authors state it can be confirmed that although the block is polluted by carbon, it only exists at the edge of the block. How is this known? Fig. 8 shows SEM, B, Si, and Fe – but no C.
- Regarding the magnetization loops, more loops than just for one sample should be shown, the same units for fields and magnetization (it is M-H, not B-H) as subsequently used in the table and discussion should be used, and the temperature at which the actual measurement took place (room temperature I assume.?) should be given.
- Units in table 1 and corresponding discussion are confusing. Magnetization has SI units of A/m, not Tesla (unless the authors mean mu0 times the magnetization, in which case that should be clearly stated. Also when making an argument about getting gradually close to theoretical value, that theoretical value should be given, along with citation or calculation.
- The authors argue that the increase in initial relative permeability is the direct result of the increase in density – this is plausible, but the argument would be much stronger by quantifying this, ideally in some figure permeability vs density.
- There are also numerous language problems and straightforward typos (e.g. muM instead of mum) in a density high enough to be distracting, and some references are incomplete.
Based on the above I cannot recommend publication of this manuscript; although an extensively revised version taking into account the points above might be reconsidered.
Author Response
Thank you very much for your meticulous guidance and help to my thesis. In response to your questions, I have made modifications within my ability. Please review it:
1.The quality of the figures is too low. While for some of the figures, this is not a big problem and the main content can still be deduced, for others (in particular Figs. 10 and 11) the legend is not readable. In Figs. 10 and 11 is is also unclear which the data correspond to the left and which to the right y-axes.
To solve this problem, I separate the two parameters in Figure 10 and Figure 11.
2.There is no motivation why SiO2 is chosen to be added to FeSiB (has SiO2 been used in literature before?), or why 6 wt% are added.
SiO2 is chosen to be added to FeSiB in many literatures,which exhibit high magnetic flux density, good soft magnetic properties and low core loss(Li, X. , et al. "Bulk amorphous powder cores with low core loss by spark-plasma sintering Fe76Si9.6B8.4P6 amorphous powder with small amounts of SiO2." Journal of Alloys and Compounds 647(2015):917-920.), 6 wt% are added is for SiO2 insulator layer forming on the surface of the alloy powder that can effectively reduce the eddy current and consequently reduce thecore loss.
3.The abstract mentions resistivity and a precision resistance tester – but in the main text I find no corresponding results. Resistivity results would certainly be useful to have at hand to support e.g. the discussion about Eddy currents.
Abstract has been modified and deleted “resistance tester”
4.Figure 2 shows particle size distribution. This is certainly useful to have – but it is completely unclear how these distributions were determined.
The particle size distribution diagram is the particle size distribution measured by laser particle size analyzer
5.Figure 3 shows XRD patterns and as the authors correctly state, there are no obvious sharp Bragg peaks characteristic of a significant crystalline volume fraction. However, there is an obvious (broad) peak between 40 and 50 deg – is this typical of these amorphous materials? What is the origin of this feature?
Figure 3 shows XRD patterns and as the authors correctly state, there are no obvious sharp Bragg peaks characteristic of a significant crystalline volume fraction. there is an obvious (broad) peak between 40 and 50 deg,There is only a halo and no distinguishable diffraction peaks of crystalline phases in XRD profile, indicating a fully amorphous structure.
6.DSC curve shown in Fig. 4: was this measured on heating or cooling? What is the most obvious peak (at around 540 C) due to and what is this arrow TC for?
More importantly, also in Fig. 4 there is an assignment of a glass transition temperature to 440 C. Looking at the figure I see no obvious feature at this temperature. To make a convincing case, maybe an inset with the immediate region enlarged could be shown, along with a brief discussion with references that only a very small feature can be expected at Tg.
The expression of DSC was changed again, and the regional enlargement of TG was added(Figure 4.)
7.How was the relative density determined? Probably by weighing and somehow determining the volume of specific samples. Still this information should be given as the density appears to be rather important for the overall conclusions.
The density of the block is accurately measured by Archimedes drainage method. The relative density of the sintered block = the measured density of the sintered sample / the theoretical density of the sintered sample, which is replaced by the powder density
8.Referencing to Fig. 8 the authors state it can be confirmed that although the block is polluted by carbon, it only exists at the edge of the block. How is this known? Fig. 8 shows SEM, B, Si, and Fe – but no C.
Deleted conclusion:”It can be confirmed that although the block is polluted by carbon, it only exists at the edge of the block. “
9.Regarding the magnetization loops, more loops than just for one sample should be shown, the same units for fields and magnetization (it is M-H, not B-H) as subsequently used in the table and discussion should be used, and the temperature at which the actual measurement took place (room temperature I assume.?) should be given.
The experimental data are added again as required, as shown in Figure 9
10.Units in table 1 and corresponding discussion are confusing. Magnetization has SI units of A/m, not Tesla (unless the authors mean mu0 times the magnetization, in which case that should be clearly stated. Also when making an argument about getting gradually close to theoretical value, that theoretical value should be given, along with citation or calculation.
Add to :Compared with ribbon samples, stronger fields should berequired for the magnetic saturation of a power disc sample due to a stronger demagnetization factor resulting from the disc shape.However, these powder discs also exhibited a typical soft magnetic loop and the magnetization saturated at the magnetic field of about 160 kA/m. The magnetiza-tion at 800A /m (B800), μe at 1kHz and Hc external magnetic fields is shown in Table 1。

Round 2
Reviewer 2 Report
I accept the modified article.
Author Response
Dear reviewer:
To"Extensive editing of English language and style required ",I have carefully revised the whole paper, and listing as following which are for your criticism.
In this study, SiO2 nanopowder was coated on a FeSiB Amorphous Powder by mechanical ball milling. After ball milling for 6h, the FeSiB / SiO2 Core-shell amorphous composite powder was obtained. The amorphous powder has thermal stability at 490℃. FeSiB / SiO2 Magnetic powder core was prepared by discharge plasma sintering (SPS). The phase composition and the magnetic properties of the magnetic particle core at sintering temperature between 420-540℃ were stud-ied. The structural characteristics of the magnetic particle core were characterized by scanning electron microscope (SEM) and X-ray diffraction (XRD). The resistivity and the magnetic proper-ties of the magnetic particle core were measured using a precision resistance tester and a vibrat-ing sample magnetometer. Findings from the study show that Fe3Si and Fe2B are the phases formed during spark plasma sintering. Higher density leads to increased high-frequency power loss. However, the magnetic permeability of the magnetic particle core fluctuates a little at the measured frequency and has excellent frequency characteristics, which is suitable for application in high-frequency components.
Revise to : Mechanical ball milling was used to coat SiO2 nanopowder on a FeSiB amorphous powder in this study. The FeSiB/SiO2 core-shell amorphous composite powder was obtained after 6 hours of ball milling. At 490 °C, the amorphous powder is thermally stable. Discharge plasma sinter-ing was used to create a FeSiB/SiO2 magnetic powder core (SPS). At a sintering temperature of 420 to 540 °C, the phase composition and magnetic characteristics of the magnetic particle core were investigated. Scanning electron microscopy (SEM) and X-ray diffraction (XRD) were used to examine the structural features of the magnetic particle core. A precision resistance tester and a vibrating sample magnetometer were used to assess the resistivity and magnetic characteristics of the magnetic particle core. Findings showed that Fe3Si and Fe2B are the phases generated dur-ing spark plasma sintering. High-frequency power loss increases as density rises. However, at the measured frequency, the magnetic permeability of the magnetic particle core changes a bit and has excellent frequency characteristics, making it appropriate for use in high-frequency components.
Fe-based amorphous alloys have been widely used in high-frequency bands because they have no grain boundary and magnetocrystalline anisotropy and have soft magnetic prop-erties such as high resistivity and low eddy current loss
Revise to : Due to their lack of grain boundary and magnetocrystalline anisotropy, as having soft magnetic properties like high resistivity and low eddy current loss, Fe-based amorphous alloys have been widely used in high-frequency bands
However, due to the high cooling rate required for the formation of the amorphous phase, iron-based amorphous alloy can only be processed by melting, which leads to limitations in its technical application
Revise to : However, Iron-based amorphous alloys, on the other hand, can only be treated by melting due to the high cooling rate required for the creation of the amorphous phase, which limits their technical application
Controlling the preparation temperature of the amorphous magnetic particle core is the key to avoiding crystallization. The preparation of an iron-based amorphous powder core still faces great challenges due to the lack of a large supercooled liquid region. Compared with traditional sintering methods, discharge plasma sintering (SPS) has significant advantages such as short duration and fast heating rate, which can inhibit the grain growth and the crystallization of the amorphous materials in the sintering process
Revise to : The key to avoiding crystallization is to keep the preparation temperature of the amorphous magnetic particle core under control. Due to the lack of a substantial supercooled liquid zone, preparing an iron-based amorphous powder core remains difficult. Compared with traditional sintering methods, discharge plasma sintering (SPS) has significant advantages like short duration and fast heating rate, which can inhibit grain growth and the crystallization of the amorphous materials during sintering
Aerosolized Fe-Si-B Amorphous Powder (purity > 99%, 40%) μ m), and SiO2 powder (purity > 99%, 50nm) are the raw material used in this experiment is.
Revise to: The raw materials used in the study include aerosolized FeSiB Amorphous Powder (purity > 99%, 40%) μ m), and SiO2 powder (purity > 99%, 50nm).
The ball milling process parameters are as follows:
the ball milling medium is 3mm stainless steel ball, the ball material ratio is 20:1-35:1, and the rotating speed is 200r/ min - 250r / min.
Revise to: The ball milling medium is a 3mm stainless steel ball with a 20:1-35:1 ball material ratio and a rotating speed of 200rpm to 250rpm.
the core-shell composite powder with Fe-Si-B as its core and a SiO2 shell is prepared.
Revise to: The core-shell composite powder is made with a FeSiB core and a SiO2 shell.
In this paper, Fe-Si-B powder-coated nano SiO2 powder with a core-shell structure was prepared by mechanical ball milling.
Revise to:Mechanical ball milling was used to generate FeSiB powder-coated nano SiO2 powder with a core-shell structure in this work.
During the continuous mechanical ball milling process, the plastic deformation caused by the collision and the extrusion resulted in the irregular shape of the composite powder and a relatively rough surface
Revise to: The plastic deformation generated by impact and extrusion during the continuous mechanical ball milling process resulted in an uneven shape of the composite powder and a rather rough surface
Fig. 2 shows the particle size distribution diagram of the core-shell powder structure.
Revise to: The particle size distribution diagram of the core-shell powder structure is shown in Figure 2.
Fig. 4 shows the DSC curve of the Fe-Si-B / SiO2 powder,
Revise to : The DSC curve of the FeSiB / SiO2 powder is shown in Figure. 4,
in which the glass transition temperature Tg is 440℃, the initial crystallization temperature TX is 480℃, the supercooled liquid region is 440℃ - 480℃, with a width of 40℃.
Revise to: with a glass transition temperature Tg of 440 °C, an initial crystallization temperature TX of 480 °C, and a supercooled liquid region of 440 °C - 480 °C, with a width of 40 ℃.
Considering the difference in temperature between the mold and the core powder in the SPS sintering, the sintering temperature is 460℃ to prevent the crystallization of the sample.
Revise to : Due to the temperature differential between the mold and the core powder in the SPS sin-tering, the sintering temperature is 460 ℃ to prevent the sample from crystallizing.
Fig. 6 shows the relative density of the sintered blocks at different sintering tempera-tures., and it shows that the density increases with an increase in the sintering tempera-ture.
Revise to : Figure 6 depicts the relative density of sintered blocks at various sintering temperatures, demonstrating that the density increases with an increase in the sintering temperature. Fig. 7 shows the SEM image of the fracture of the magnetic particle core sintered at 460℃.
Fig. 7 shows the SEM image of the fracture of the magnetic particle core sintered at 460℃.
Revise to : The SEM image of the fracture of the magnetic particle core sintered at 460 ℃ is shown in Figure. 7.
It can be seen that there are many visible traces of the liquid layer on the surface of the particle and between the particles, and the SiO2 nanopowder melts and fills the gap in the Fe-Si-B particles. This shows that in the SPS process, the plasma formation is highly exothermic, so the temperature at the surface edge can even rise to thousands of degrees Celsius
Revise to : There are many apparent remnants of the liquid layer on the surface of the particle and between the particles, and the SiO2 nanopowder melts and fills the gap in the FeSiB particles. This shows that plasma formation is highly exothermic in the SPS process, hence, the temperature at the edge of the surface can rise to thousands of degrees Celsius
EDS of polished surface of Fe-Si-B/SiO2 Magnetic particle core is shown in Figure 9.
Revise to : Figure 9 shows the EDS spectrum of the polished surface of the FeSiB / SiO2 powder core.
In the high-frequency range, the maximum loss is the holding time of 5min. Sintering times
Revise to : The maximum loss in the high-frequency spectrum is a 5-minute holding time.
The evolution of the magnetic properties can be explained by the density and resistivity characteristics of the magnetic particle core.
Revise to : The density and resistivity characteristics of the magnetic particle core help explain the evolution of the magnetic properties.
eddy current loss is the main contribution to the total magnetic loss in the magnetic parti-cle core.
Revise to : The eddy current loss is directly proportional to the square of the applied frequency and inversely proportional to the compact resistivity.
Considering the XRD,
Revise to : Given that the XRD
Therefore, it can be concluded that density is the main reason for the change in permeability.
Revise to : As a result, it can be inferred that the main cause of the change in permeability is density.
The increase in initial relative permeability is the direct result of the increase in density.
Revise to : The increase in density causes a direct rise in the initial relative permeability.
The density is low and the porosity of the sample is large, resulting in decreased magnetic permeability.
Revise to : The sample's density is low, and its porosity is high, resulting in poor magnetic permeability.
It can be observed that the permeability is almost constant in the whole frequency range during the 5 min holding time.
Revise to : During the 5-minute holding period, the permeability is essentially consistent over the whole frequency range.
In addition,
Revise to : Furthermore,
but when the sintering temperature is 540℃, the iron loss decreases.
Revise to : but the iron loss decreases when the sintering temperature is 540 ℃.
so it increases
Revise to : which increases
and the bulk is dominated by nanocrystals.
Revise to : and nanocrytals dominate the bulk.
In this process, the density of the bulk increases, and the micro stress in the powder is re-leased during the sintering process.
Revise to : During the sintering process, the density of the bulk increases, and the micro tension in the powder is released.
The core-shell composite powder of the Fe-Si-B coated nano SiO2 nanopowder was prepared by mechanical ball milling. The width of the supercooled liquid region of the composite powder is up to 40℃, and the amorphous alloy has a strong amorphous forming ability.
Revise to : Mechanical ball milling was used to prepare the core-shell composite powder of the FeSiB coated nano SiO2 nanopowder. The width of the supercooled liquid region of the composite powder is up to 40 ℃, and the amorphous alloy has a strong amorphous forming ability.
The Fe-Si-B / SiO2 bulk amorphous nanocrystalline magnetic materials were prepared by spark plasma sintering. When the sintering pressure was 40 MPa, the sintering tempera-ture range was 420-540℃, with a holding time of 1 min, the prepared bulk Fe-Si-B particles (core) were well separated and insulated by the SiO2 (shell) intergranular layer in the core of the powder, and the density reached 98.93%. The block crystallizes at about 460℃, and its crystalline phases are Fe3Si and Fe2B.
Revise to : Spark plasma sintering was used to prepare bulk amorphous nanocrystalline magnetic materials made of FeSiB and SiO2. The manufactured bulk FeSiB particles (core) were well separated and insulated by the intergranular layer of the SiO2 (shell) in the core of the powder when the sintering pressure was 40 MPa and the sintering temperature range was 420-540 °C, with a holding period of 1 min, and the density reached 98.93%. Fe3Si and Fe2B are the crystalline phases of the block, which crystallize at around 460 °C.
When the sintering and holding times are increased, there is an improvement in the density, maximum relative permeability, and magnetic loss of the Fe-Si-B / SiO2 sintered block. The magnetic permeability of the Fe-Si-B / SiO2 sintered block is stable in the high-frequency region. The magnetic properties are the best when the sintering temperature is 420℃ for 2min, with excellent soft magnetic properties.
Revise to : Increasing the sintering and holding times, results in an improvement in the density, maximum relative permeability, and the magnetic loss of the FeSiB / SiO2 sintered block. The magnetic permeability of the FeSiB / SiO2 sintered block is stable in the high-frequency region. The magnetic properties are the best when the sintering temperature is 420 ℃ for 2min, with excellent soft magnetic properties.

Reviewer 3 Report
The resubmitted manuscript is improved, with the inclusion of additional magnetization data, the modifications of Figs. 10 and 11, and a couple of clarifications in the text. However, while I am satisfied with the authors’ response of some of the concerns raised in my previous report, I am not convinced by the response to others. For several points, the response as such if fine but I cannot discern corresponding changes in the manuscript (4,7) or the change in the manuscript lacks inclusion of a reference given in the response (2). For others, answers to the question or part of the question are missing (e.g. on 9 the temperature at which the magnetization measurements were carried out).
Most concerning, are the replies to points 3 and 8:
- In response to the inquiry about electrical resistance mentioned in abstract and with no corresponding data shown in the paper, authors removed “precision resistance tester” from abstract, while keeping the also mentioned “resistivity” there. The sentence as now in the abstract implies that resistivity was measured with a vibrating sample magnetometer, which is not possible (and there are still no resistivity data shown in paper with no response to their in my previous report suggested desirability).
- In response to the question how the authors know about where carbon pollution occurs (given no map for carbon shown in the EDX), the authors just deleted the conclusion where carbon pollution occurs from the manuscript (which seems to indicate they have no answer). This leads to a new problem, however, as the unchanged start of the paragraph about the EDX clearly states that the potential pollution by carbon was the reason the EDX microanalysis was performed.
In, summary, despite some improvements I am still not able to recommend the present version.
Author Response
Dear Reviewer:
Thank you for your pertinent advice1
1).In response to the inquiry about electrical resistance mentioned in abstract and with no corresponding data shown in the paper, authors removed “precision resistance tester” from abstract, while keeping the also mentioned “resistivity” there. The sentence as now in the abstract implies that resistivity was measured with a vibrating sample magnetometer, which is not possible (and there are still no resistivity data shown in paper with no response to their in my previous report suggested desirability).
For the resistance query mentioned in the summary, I give the corresponding data in the paper, as shown in Table 1, and the "precision resistance tester" is restored in the summary .The resistivity of the FeSiB/ SiO2 decreases with the increase of sintering temperature, but it is sig-nificantly higher than that of the comparative amorphous belt sample I (137.1µΩ•cm), which is because the SiO2 insulating layer hinders the electron movement in Fe-Si-B particles and achieves good insulation effect.
2).In response to the question how the authors know about where carbon pollution occurs (given no map for carbon shown in the EDX), the authors just deleted the conclusion where carbon pollution occurs from the manuscript (which seems to indicate they have no answer). This leads to a new problem, however, as the unchanged start of the paragraph about the EDX clearly states that the potential pollution by carbon was the reason the EDX microanalysis was performed.
In order to confirm the carbon contamination of the dies and punches from SPS equipment, EDX microanalysis was performed on the edge of the compact slice obtained by SPS at 460℃. As shown in Figure 8. It is found that there is a large amount of carbon on the surface of the magnetic particle core, and the diffusion depth of carbon is estimated to be about 2-4 μm .
3).In addition, I have carefully polished and revised the language expression and grammar of the full text. Listed below:
1.In this study, SiO2 nanopowder was coated on a FeSiB Amorphous Powder by mechanical ball milling. After ball milling for 6h, the FeSiB / SiO2 Core-shell amorphous composite powder was obtained. The amorphous powder has thermal stability at 490℃. FeSiB / SiO2 Magnetic powder core was prepared by discharge plasma sintering (SPS). The phase composition and the magnetic properties of the magnetic particle core at sintering temperature between 420-540℃ were stud-ied. The structural characteristics of the magnetic particle core were characterized by scanning electron microscope (SEM) and X-ray diffraction (XRD). The resistivity and the magnetic proper-ties of the magnetic particle core were measured using a precision resistance tester and a vibrat-ing sample magnetometer. Findings from the study show that Fe3Si and Fe2B are the phases formed during spark plasma sintering. Higher density leads to increased high-frequency power loss. However, the magnetic permeability of the magnetic particle core fluctuates a little at the measured frequency and has excellent frequency characteristics, which is suitable for application in high-frequency components.
Revise to : Mechanical ball milling was used to coat SiO2 nanopowder on a FeSiB amorphous powder in this study. The FeSiB/SiO2 core-shell amorphous composite powder was obtained after 6 hours of ball milling. At 490 °C, the amorphous powder is thermally stable. Discharge plasma sinter-ing was used to create a FeSiB/SiO2 magnetic powder core (SPS). At a sintering temperature of 420 to 540 °C, the phase composition and magnetic characteristics of the magnetic particle core were investigated. Scanning electron microscopy (SEM) and X-ray diffraction (XRD) were used to examine the structural features of the magnetic particle core. A precision resistance tester and a vibrating sample magnetometer were used to assess the resistivity and magnetic characteristics of the magnetic particle core. Findings showed that Fe3Si and Fe2B are the phases generated dur-ing spark plasma sintering. High-frequency power loss increases as density rises. However, at the measured frequency, the magnetic permeability of the magnetic particle core changes a bit and has excellent frequency characteristics, making it appropriate for use in high-frequency components.
2.Fe-based amorphous alloys have been widely used in high-frequency bands because they have no grain boundary and magnetocrystalline anisotropy and have soft magnetic prop-erties such as high resistivity and low eddy current loss
Revise to : Due to their lack of grain boundary and magnetocrystalline anisotropy, as having soft magnetic properties like high resistivity and low eddy current loss, Fe-based amorphous alloys have been widely used in high-frequency bands
3.However, due to the high cooling rate required for the formation of the amorphous phase, iron-based amorphous alloy can only be processed by melting, which leads to limitations in its technical application
Revise to : However, Iron-based amorphous alloys, on the other hand, can only be treated by melting due to the high cooling rate required for the creation of the amorphous phase, which limits their technical application
4.Controlling the preparation temperature of the amorphous magnetic particle core is the key to avoiding crystallization. The preparation of an iron-based amorphous powder core still faces great challenges due to the lack of a large supercooled liquid region. Compared with traditional sintering methods, discharge plasma sintering (SPS) has significant advantages such as short duration and fast heating rate, which can inhibit the grain growth and the crystallization of the amorphous materials in the sintering process
Revise to : The key to avoiding crystallization is to keep the preparation temperature of the amorphous magnetic particle core under control. Due to the lack of a substantial supercooled liquid zone, preparing an iron-based amorphous powder core remains difficult. Compared with traditional sintering methods, discharge plasma sintering (SPS) has significant advantages like short duration and fast heating rate, which can inhibit grain growth and the crystallization of the amorphous materials during sintering
5.Aerosolized Fe-Si-B Amorphous Powder (purity > 99%, 40%) μ m), and SiO2 powder (purity > 99%, 50nm) are the raw material used in this experiment is.
Revise to: The raw materials used in the study include aerosolized FeSiB Amorphous Powder (purity > 99%, 40%) μ m), and SiO2 powder (purity > 99%, 50nm).
The ball milling process parameters are as follows:
6.the ball milling medium is 3mm stainless steel ball, the ball material ratio is 20:1-35:1, and the rotating speed is 200r/ min - 250r / min.
Revise to: The ball milling medium is a 3mm stainless steel ball with a 20:1-35:1 ball material ratio and a rotating speed of 200rpm to 250rpm.
7.the core-shell composite powder with Fe-Si-B as its core and a SiO2 shell is prepared.
Revise to: The core-shell composite powder is made with a FeSiB core and a SiO2 shell.
8.In this paper, Fe-Si-B powder-coated nano SiO2 powder with a core-shell structure was prepared by mechanical ball milling.
Revise to:Mechanical ball milling was used to generate FeSiB powder-coated nano SiO2 powder with a core-shell structure in this work.
9.During the continuous mechanical ball milling process, the plastic deformation caused by the collision and the extrusion resulted in the irregular shape of the composite powder and a relatively rough surface
Revise to: The plastic deformation generated by impact and extrusion during the continuous mechanical ball milling process resulted in an uneven shape of the composite powder and a rather rough surface
10.Fig. 2 shows the particle size distribution diagram of the core-shell powder structure.
Revise to: The particle size distribution diagram of the core-shell powder structure is shown in Figure 2.
11.Fig. 4 shows the DSC curve of the Fe-Si-B / SiO2 powder,
Revise to : The DSC curve of the FeSiB / SiO2 powder is shown in Figure. 4,
12.in which the glass transition temperature Tg is 440℃, the initial crystallization temperature TX is 480℃, the supercooled liquid region is 440℃ - 480℃, with a width of 40℃.
Revise to: with a glass transition temperature Tg of 440 °C, an initial crystallization temperature TX of 480 °C, and a supercooled liquid region of 440 °C - 480 °C, with a width of 40 ℃.
13.Considering the difference in temperature between the mold and the core powder in the SPS sintering, the sintering temperature is 460℃ to prevent the crystallization of the sample.
Revise to : Due to the temperature differential between the mold and the core powder in the SPS sin-tering, the sintering temperature is 460 ℃ to prevent the sample from crystallizing.
14.Fig. 6 shows the relative density of the sintered blocks at different sintering tempera-tures., and it shows that the density increases with an increase in the sintering tempera-ture.
Revise to : Figure 6 depicts the relative density of sintered blocks at various sintering temperatures, demonstrating that the density increases with an increase in the sintering temperature. Fig. 7 shows the SEM image of the fracture of the magnetic particle core sintered at 460℃.
15.Fig. 7 shows the SEM image of the fracture of the magnetic particle core sintered at 460℃.
Revise to : The SEM image of the fracture of the magnetic particle core sintered at 460 ℃ is shown in Figure. 7.
16.It can be seen that there are many visible traces of the liquid layer on the surface of the particle and between the particles, and the SiO2 nanopowder melts and fills the gap in the Fe-Si-B particles. This shows that in the SPS process, the plasma formation is highly exothermic, so the temperature at the surface edge can even rise to thousands of degrees Celsius
Revise to : There are many apparent remnants of the liquid layer on the surface of the particle and between the particles, and the SiO2 nanopowder melts and fills the gap in the FeSiB particles. This shows that plasma formation is highly exothermic in the SPS process, hence, the temperature at the edge of the surface can rise to thousands of degrees Celsius
17.EDS of polished surface of Fe-Si-B/SiO2 Magnetic particle core is shown in Figure 9.
Revise to : Figure 9 shows the EDS spectrum of the polished surface of the FeSiB / SiO2 powder core.
18.In the high-frequency range, the maximum loss is the holding time of 5min. Sintering times
Revise to : The maximum loss in the high-frequency spectrum is a 5-minute holding time.
19.The evolution of the magnetic properties can be explained by the density and resistivity characteristics of the magnetic particle core.
Revise to : The density and resistivity characteristics of the magnetic particle core help explain the evolution of the magnetic properties.
20.eddy current loss is the main contribution to the total magnetic loss in the magnetic parti-cle core.
Revise to : The eddy current loss is directly proportional to the square of the applied frequency and inversely proportional to the compact resistivity.
21.Considering the XRD,
Revise to : Given that the XRD
22.Therefore, it can be concluded that density is the main reason for the change in permeability.
Revise to : As a result, it can be inferred that the main cause of the change in permeability is density.
23.The increase in initial relative permeability is the direct result of the increase in density.
Revise to : The increase in density causes a direct rise in the initial relative permeability.
24.The density is low and the porosity of the sample is large, resulting in decreased magnetic permeability.
Revise to : The sample's density is low, and its porosity is high, resulting in poor magnetic permeability.
25.It can be observed that the permeability is almost constant in the whole frequency range during the 5 min holding time.
Revise to : During the 5-minute holding period, the permeability is essentially consistent over the whole frequency range.
26.In addition,
Revise to : Furthermore,
27.but when the sintering temperature is 540℃, the iron loss decreases.
Revise to : but the iron loss decreases when the sintering temperature is 540 ℃.
28.so it increases
Revise to : which increases
29.and the bulk is dominated by nanocrystals.
Revise to : and nanocrytals dominate the bulk.
30.In this process, the density of the bulk increases, and the micro stress in the powder is re-leased during the sintering process.
Revise to : During the sintering process, the density of the bulk increases, and the micro tension in the powder is released.
31.The core-shell composite powder of the Fe-Si-B coated nano SiO2 nanopowder was prepared by mechanical ball milling. The width of the supercooled liquid region of the composite powder is up to 40℃, and the amorphous alloy has a strong amorphous forming ability.
Revise to : Mechanical ball milling was used to prepare the core-shell composite powder of the FeSiB coated nano SiO2 nanopowder. The width of the supercooled liquid region of the composite powder is up to 40 ℃, and the amorphous alloy has a strong amorphous forming ability.
32.The Fe-Si-B / SiO2 bulk amorphous nanocrystalline magnetic materials were prepared by spark plasma sintering. When the sintering pressure was 40 MPa, the sintering tempera-ture range was 420-540℃, with a holding time of 1 min, the prepared bulk Fe-Si-B particles (core) were well separated and insulated by the SiO2 (shell) intergranular layer in the core of the powder, and the density reached 98.93%. The block crystallizes at about 460℃, and its crystalline phases are Fe3Si and Fe2B.
Revise to : Spark plasma sintering was used to prepare bulk amorphous nanocrystalline magnetic materials made of FeSiB and SiO2. The manufactured bulk FeSiB particles (core) were well separated and insulated by the intergranular layer of the SiO2 (shell) in the core of the powder when the sintering pressure was 40 MPa and the sintering temperature range was 420-540 °C, with a holding period of 1 min, and the density reached 98.93%. Fe3Si and Fe2B are the crystalline phases of the block, which crystallize at around 460 °C.
33.When the sintering and holding times are increased, there is an improvement in the density, maximum relative permeability, and magnetic loss of the Fe-Si-B / SiO2 sintered block. The magnetic permeability of the Fe-Si-B / SiO2 sintered block is stable in the high-frequency region. The magnetic properties are the best when the sintering temperature is 420℃ for 2min, with excellent soft magnetic properties.
Revise to : Increasing the sintering and holding times, results in an improvement in the density, maximum relative permeability, and the magnetic loss of the FeSiB / SiO2 sintered block. The magnetic permeability of the FeSiB / SiO2 sintered block is stable in the high-frequency region. The magnetic properties are the best when the sintering temperature is 420 ℃ for 2min, with excellent soft magnetic properties.
